# G-CutMix: A CutMix-based graph data augmentation method for bot detection in social networks

**Yan Li**[1], **Shuhao Shi**[2], **Xiaofeng Guo**[2*], **Chunhua Zhou**[2], **Qian Hu**[2]

**1** WuXi University, Wuxi, Jiangsu, China, **2** Zhengzhou Information Science and Technology Institue, Zhengzhou, Henan, China

☯ These authors contributed equally to this work.

* yanli@cwxu.edu.cn

**Data availability statement:** All datasets used in this study are publicly accessible without

## Abstract

The CutMix technique is a sophisticated approach for augmenting data in order to train neural network-based image classifiers. Essentially, it involves cutting out a portion of a random image and pasting it into the same location as another image. However, because of the irregularity of graph data, CutMix cannot be directly applied to graph learning. Our paper introduces G-CutMix, a CutMix-based data augmentation approach that we designed specifically for bot detection in social media networks. G-CutMix involves conducting CutMix operations between the original graph and a shuffled graph, which precedes the graph convolution process. The outputs of the graph convolution are then strategically merged with the user representations from both the original and shuffled graphs. Our proposed G-CutMix not only leverages the power of graph convolutions but also introduces a layer of complexity that mimics real-world scenarios where bot behavior can be subtle and varied, making G-CutMix a formidable tool in the arsenal against bot detection. Our experiments confirm that our approach can consistently enhance the performance of bot detection across various GNN architectures, including Graph Convolutional Networks, GraphSAGE, and Graph Attention Networks.

## Introduction

Graph neural networks (GNNs) have attained remarkable accuracy in detecting social network bots due to their ability to model interactions between accounts. However, obtaining a significant number of high-quality account annotations in practical scenarios is challenging, and the number of samples available for model training may be limited. Consequently, a well-trained GNN might end up learning random errors or noise rather than the true data distribution [1], which is suboptimal.

Social bot detection techniques are currently classified into three categories: feature-based methods, graph-based methods, and deep learning methods. Feature-based approaches have been in use since the earliest efforts in social bots detection. This approach involves extracting and designing features [2] from users' metadata, such as Twitter content [3], user

restrictions: MGTAB Dataset: GitHub repository: https://github.com/GraphDetec/MGTAB (accessed: [Date, e.g., May 2024]) TwiBot-20 Dataset: GitHub repository: https://github.com/BunsenFeng/TwiBot-20 Cresci-15 Dataset: Official portal: Institute of Informatics and Telematics, National Research Council (CNR), Italy http://mib.projects.iit.cnr.it/dataset.html.

**Funding:** The National Natural Science Foundation of China 61872448, 62002387. The funders had no role in study design, data collection and analysis, decision to publish, or preparation of the manuscript.

**Competing interests:** The authors have declared that no competing interests exist.

data [4], features [5], timeliness, language, and emotions extracted from friends and neighbors' information. Traditional classifiers are then employed for robot detection. However, social bots can modify their registration information based on the features designed for detection, thereby avoiding feature-based detection methods [6,7].

The graph-based approach utilizes the graph structure of the social network to anticipate the labels of each remaining node. These techniques do not require a significant amount of text and media data that are highly dependent on language. Based on the core algorithms, they can be broadly categorized into two types: the random walk method [8–10] and the Loopy Belief Propagation (LBP) method [11,12]. Both approaches start with a set of labeled nodes and use semi-supervised learning to predict the labels of unknown nodes. The effective utilization of internal data on social platforms enhances its detection capability significantly higher than the aforementioned method.

The deep learning-based detection method primarily relies on graph representation learning or GNN model, which simultaneously utilizes the user's attribute and structure information to enhance the detection performance. TrustGCN [13] employs the "friend request-reply" relationship to form a social graph and combines the defensive concepts based on the social graph with GNN to improve the robustness of adversarial attacks. Bot2vec [14] is an improved social robot detection algorithm that is based on Node2vec. It combines a clustering technique with a graph representation learning algorithm for social bot detection. The proposed BotRGCN [15] uses graph convolutional networks to analyze Twitter social graphs, leveraging user attributes and interactions for detecting social bots. SATAR [16] is a self-supervised representation method that leverages user tweets, metadata, and connections to analyze Twitter users.

Recently, data augmentation has proved to be highly effective in convolutional neural networks (CNNs). However, data augmentation in graph neural networks (GNNs) is still sparsely studied. For node classification tasks, Rong et al. [17] proposed a data augmentation technique named DropEdge, which basically involves randomly removing a certain number of edges from the input graph in each training stage. Wang et al. [18] developed MixupForGraph, which starts with standard feed-forward operations and then applies Mixup graph convolutions to nodes in the second stage. Han et al. [19] introduced G-Mixup, a method that interpolates between graphons from different graph classes to enhance graph classification.

We present a novel approach for augmenting data in bot detection using graph learning. Initially, we shuffle the user relationship graph to obtain its isomorphic graph. Next, we cutmix the node features and labels of the two graphs in order. The resultant mixed features are then fed into the graph convolution. The local aggregation-based features produced by the graph convolution are added to the original features of the two views. The outcome is a new layer of features for the two views.

Our cutmix methods can be integrated into well-known GNNs. We assess the efficiency of our approach on real-world bot detection datasets, including Cresci-15 [20], Twibot-20 [21], and MGTAB [22]. In summary, our contribution is threefold:

- We introduce a novel and versatile data augmentation technique for graph learning based on CutMix. Compared with previous data enhancement methods, our method is more effective.
- We have extended our method to heterogeneous graphs and demonstrated that integrating information from various relationships between users can significantly enhance the performance of bot detection graph representation learning.
- Our approach has undergone extensive experimentation on real-world bot detection datasets, which has shown its high efficacy. Compared to earlier data augmentation

methods, our approach shows a substantial enhancement in both accuracy and F1-score for bot detection.

## Related work

### Graph neural network

GNNs process non-Euclidean graph data by generating node-level feature representations through message passing. At layer $l + 1$, node features are obtained by aggregating information from their 1-hop neighbors at layer $l$:

$$\boldsymbol{h}^{(l+1)} = \sigma\left(AGGREGATE\left(\boldsymbol{h}^{(l)}\boldsymbol{W}, \mathcal{A}\right)\right)$$

Where $\mathcal{A}$ is the adjacency matrix, and $\boldsymbol{W}$ represents the learnable parameter matrix, and $\sigma$ represents the activation function. *AGGREGATE* represents the aggregation function that aggregates the hidden representations of neighboring nodes.

### GNN-based bot detection

Graph-based methods have proven highly effective in social bot detection [21,22]. START [23] develops a method for learning Twitter user representations, which is then used to refine bot detection. GNNs have recently made significant advances in this area, modeling accounts as nodes and relationships such as friends and followers as edges. Alhosseini et al. [24] were pioneers in applying Graph Convolutional Networks to social bot detection, utilizing account interactions effectively. More recently, NDE-GNN [25] achieves social bot detection by integrating hypergraph-based higher-order neighborhood representations with differential feature enhancement techniques. Shi et al. [26] introduced RF-GNN, which enhances detection accuracy through ensemble learning. However, RF-GNN's use of averaging to combine base classifiers can reduce effectiveness by not fully capturing the performance variations among different classifiers. Li et al. proposed BotCL [27], a graph contrastive learning-based social bot detection model leveraging multi-view data augmentation to integrate semantic, attribute, and structural features, yet it faces hyperparameter sensitivity and incomplete utilization of heterogeneous social relationships.

### Data augmentation

Data augmentation is essential for enhancing model performance, as it modifies the input data to create a more robust learning environment. For instance, in image classification, data augmentation techniques like horizontal flipping and random erasing have been demonstrated to improve performance. Mixup [28] is a powerful data augmentation technique for image classification, involving training a neural network with convex combinations of pairs of images and their labels. Recently, several mixup-based methods have been proposed for graph data to enhance the performance of models. G-Mixup [29] is a data augmentation technique for graph classification that generates a new graph by using a linear combination of two graphs. MixupForGraph [18] combines two graphs in a manner that preserves the graph structure. GraphMix [36] integrates a graph neural network with traditional neural network architectures, using advanced techniques to improve performance.

CutMix [30] represents a unique data augmentation approach that randomly mixes the input data and labels of two samples, training models on the combined data to encourage learning from diverse parts of multiple examples—a strategy more robust than traditional Mixup. Building on this, we developed G-CutMix, a CutMix-based graph data augmentation

method designed to enhance GNN performance. Unlike prior Mixup-based graph augmentation methods (e.g., MixupForGraph's global structure blending), G-CutMix employs local subgraph replacement rather than global interpolation, striking a balance between data diversity and the preservation of local graph structures and topological relationships. This mechanism gives G-CutMix distinct advantages in tasks relying on local information, such as node classification or molecular property prediction. In contrast, MixupForGraph's global mixing nature may be more suitable for tasks insensitive to fine-grained local structures. With its physical interpretability and ability to retain local structures, G-CutMix emerges as a superior choice for graph data augmentation—addressing the unique challenges of structural data while enhancing model generalization.

## Proposed method

### Background and motivation

Data augmentation is a straightforward yet effective method to enhance neural network training by expanding the diversity of the training dataset. Mixup-based graph enhancement methods have been proposed, significantly improving the ability of graph representation learning. CutMix and its variants use region-level cut and paste hybrid techniques to force the neural network to pay more attention to the global context of the image rather than just local information, thereby maintaining continuity of information compared to mixup. Unlike MixupForGraph, which linearly interpolates node features, G-CutMix employs region-level feature swapping using a binary mask M (Eq 1). This ensures that local structural patterns (e.g., community-specific interactions) are preserved during augmentation, whereas linear interpolation may blur such patterns. For example, in social graphs, bots often exhibit localized behavioral anomalies (e.g., sudden spikes in follower requests). By retaining intact feature regions, G-CutMix helps GNNs capture these subtle signals more effectively.

In our work, we perform bot detection based on node classification and propose a graph augmentation method named G-CutMix, utilizing a CutMix module to improve the performance of bot detection. The overall framework of G-CutMix is depicted in Fig 1 and comprises of three modules: Graph Shuffle Module, Node CutMix Module, and Attribute Connection Module.

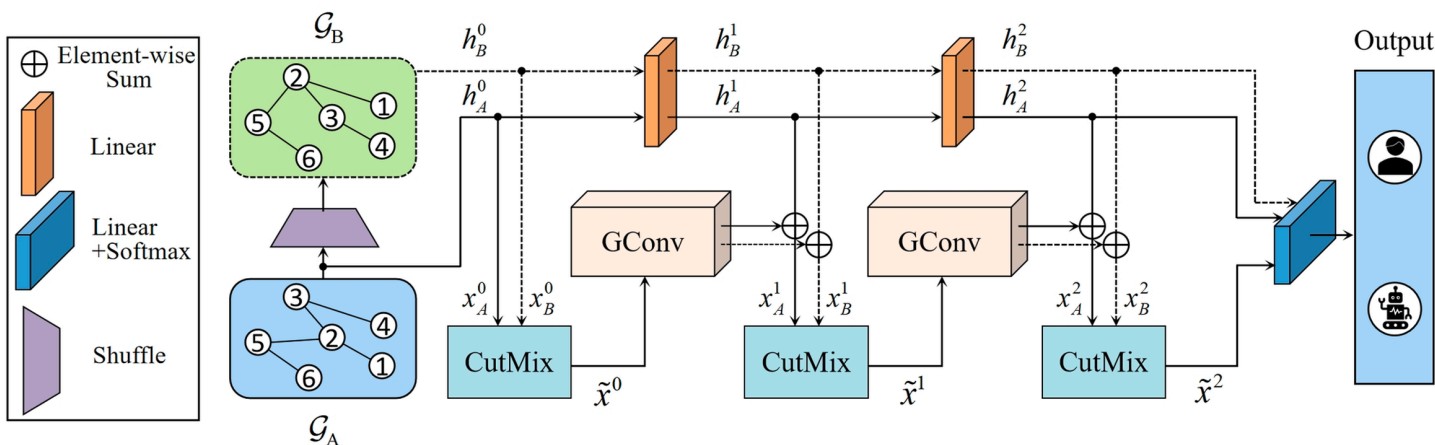

**Fig 1. The procedure for training with G-CutMix.** $\mathcal{G}_A$ and $\mathcal{G}_B$ is isomorphic graph.

## Graph shuffle module

The objective of node shuffling is to disturb the arrangement of nodes and their associated edges within the graph, resulting in the isomorphic graph $\mathcal{G}_B$, which is based on the original graph $\mathcal{G}_A$. The definition of an isomorphic graph can be found in **Definition 1**. To accomplish this, the node ID $I(\mathcal{G}_A)$ in $\mathcal{G}_A$ must be shuffled to obtain the shuffled ID $\widetilde{I}(\mathcal{G}_A)$. The nodes within $\mathcal{G}_A$ should be rearranged according to the order of $\widetilde{I}(\mathcal{G}_A)$. The ratio of node shuffling can be denoted by the variable $\lambda$, $\lambda \in [0, 1]$.

**Definition 1.** *Isomorphic graph: Given two graphs $\mathcal{G}_A = (\mathcal{V}_A, \mathcal{E}_A)$ and $\mathcal{G}_A = (\mathcal{V}_B, \mathcal{E}_B)$, if there exists a bijection m: $\mathcal{V}_A \to \mathcal{V}_B$ such that for all $v, u \in \mathcal{V}_A, u, v \in \mathcal{E}_A$ is equivalent to $m(u)m(v) \in \mathcal{E}_B$, then $\mathcal{G}_A$ and $\mathcal{G}_B$ are isomorphic.*

## Node CutMix module

Sample pairs are created by selecting corresponding numbered nodes from $\mathcal{G}_A$ and $\mathcal{G}_B$. For instance, the sample pair constructed from the *i*-th node in both graphs is represented as $(\boldsymbol{h}_{A,i}, \boldsymbol{h}_{B,i})$, $1 \leq i \leq N$, where $N$ denotes the number of nodes within the graph. The Node Cut-Mix Module is depicted in Fig 2.

CutMix the representations of sample pairs according to Eq 1:

$$\widetilde{\boldsymbol{x}}_i^l = \boldsymbol{M}^l \odot \boldsymbol{x}_{A,i}^l + (1 - \boldsymbol{M}^l) \odot \boldsymbol{x}_{B,i}^l \tag{1}$$

here $\boldsymbol{M} \in \{0, 1\}^H$ represents a binary mask of length $H$, where each element indicates whether to drop out or retain the corresponding entry from two vectors. $\alpha \in [0, 1]$ is the hyperparameter that controls the ratio of 1 in mask. $\boldsymbol{x}_{A,i}^0 = \boldsymbol{h}_{A,i}^0$ and $\boldsymbol{x}_{B,i}^0 = \boldsymbol{h}_{B,i}^0$. $X_A^l$ and $X_B^l$ pass through the Graph CutMix Module to get $\widetilde{X}^l$, $\widetilde{X}^l = \left[\widetilde{\boldsymbol{x}}_i^l\right]$.

## Attribute connection module

The Attribute Connection Module comprises a full connection layer and is utilized to preserve the original attributes of the graph. The output obtained by $\widetilde{X}^l$ via the graph convolution layer (GConv) and the initial representations are obtained by adding features $\boldsymbol{h}_{A,i}^l$ and $\boldsymbol{h}_{B,i}^l$ through the linear layer, resulting in obtaining $\boldsymbol{x}_{A,i}^{l+1}$ and $\boldsymbol{x}_{B,i}^{l+1}$:

$$\boldsymbol{x}_{A,i}^{l+1} = \sigma\left(AGGREGATE(\widetilde{\boldsymbol{x}}^l \boldsymbol{W}), \mathcal{A}_{\mathcal{G}_A}\right) + \boldsymbol{h}_{A,i}^l \tag{2}$$

$$\boldsymbol{x}_{B,i}^{l+1} = \sigma\left(AGGREGATE(\widetilde{\boldsymbol{x}}^l \boldsymbol{W}), \mathcal{A}_{\mathcal{G}_B}\right) + \boldsymbol{h}_{B,i}^l \tag{3}$$

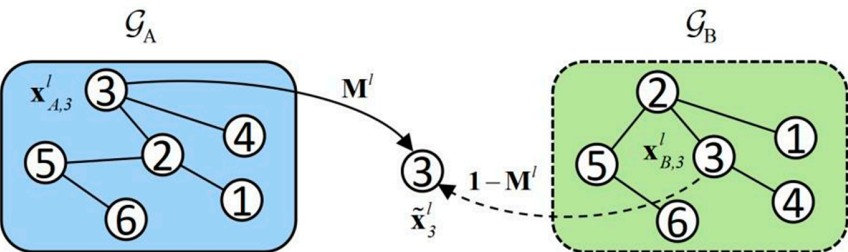

**Fig 2. Node Cutmix module in our proposed G-CutMix.**

Afterward, the resulting $x_{A,i}^{l+1}$ and $x_{B,i}^{l+1}$ are fed into the Node CutMix Module in the subsequent layer. Once CutMix is employed to enhance the training distribution features, corresponding labels should be mixed while computing the loss function:

$$\widetilde{y} = \alpha y_A + (1 - \alpha) y_B \tag{4}$$

## Experiment

### Dataset

We assess the performance of Twitter bot detection models on three datasets that possess graph structures:

- Cresci-15 [20] is a dataset with 5,301 users labeled as either genuine or automated accounts, and it includes details on their follower and friend relationships.
- Twibot-20 [21] is a dataset comprising 229,580 users and 227,979 edges, with 11,826 accounts labeled as either genuine or automated. It includes detailed information on follower and friend relationships among these users.
- MGTAB [22] is a comprehensive dataset designed for machine account detection, featuring over 1.5 million users and 130 million tweets. It includes data on seven types of relationships between these users and labels 10,199 accounts as either genuine or bots. The dataset details seven types of relationships among users and labels 10,199 accounts as either genuine or bots.

We build user social graphs using labeled users from each dataset. For MGTAB, we incorporate 20 high-information-gain user attributes and 768-dimensional tweet features from BERT. In Twibot-20, we use 17 user attributes, 768-dimensional description features from BERT, and tweet features. For Cresci-15, we use 6 user attributes, 768-dimensional description features from BERT, and tweet features. The details of these datasets are presented in Table 1. For all datasets, we partition the data randomly into training, validation, and testing sets in a 0.5:1:8.5 ratio.

### Baseline methods

To evaluate the effectiveness of our proposed method, we compare it against commonly used GNN methods, including:

**GCN**: [31] is a method used for semi-supervised learning on graph-structured data, focusing on localized graph convolution.

**SAGE**: [32] comprises sampling and aggregation. Initially, it selects neighboring nodes based on their connections and then merges information from these neighbors through a series of multi-layer aggregation functions. The fused information is used to predict the node label.

**GAT**: [33] utilizes an attention mechanism to assign weights to node neighborhoods. This adaptive weighting of neighbors enhances the performance of graph neural networks.

**Table 1**. Statistics of bot detection datasets used in the paper.

| Dataset | Nodes | Edges | Features | Relations |
|---------|-------|-------|----------|-----------|
| Cresci-15 | 5,301 | 14,220 | 1,542 | 2 |
| Twibot-20 | 11,826 | 15,434 | 1,553 | 2 |
| MGTAB | 10,199 | 1,700,108 | 784 | 7 |

**JK-Nets**: [34] mitigates over-smoothing concerns in deep GNNs by harnessing varying neighborhood scopes for flexible aggregation.

**LA-GCN**: [35] introduces a layer-wise attention mechanism, enabling the network to discern the significance of each layer's output in the ultimate node representation.

**RF-GNN**: [26] is a method that randomly deletes a portion of edges from the input graph at each training stage.

**DropEdge**: [17] is a technique that randomly removes a certain number of edges from the input graph during each training stage.

**MixupForGraph**: [18] is a two-stage method for GNNs. It first performs standard feed-forward operations, then incorporates Mixup-based techniques to blend information from different nodes during each layer of the network.

## Configurations

All of our models are constructed with two layers of graph convolution layers with ReLU activation and a dropout rate of 0.5. We employ the Cross-Entropy loss function and the Adam optimizer for model optimization, setting the learning rate to 1e-3 and a weight decay to 5e-4. For the Cresci-15 [20], Twibot-20 [21], and MGTAB [22] datasets, we utilize followers and friends user relationships, respectively. We follow [21] to extract and construct user feature attributes. We set $\alpha$ (the hyperparameter for the adaptive threshold) to 0.4 and train the model for 200 epochs. Attention heads in GAT and RGAT are configured to 4.

## Main results

In this section, we investigate the performance of G-CutMix using follower and friend relationships in bot detection tasks, respectively. In our experiments, we created social relationship graphs using followers and friends relationships and then utilized graph neural network models to detect machine accounts. Each baseline was executed five times with varying initializations to mitigate randomness. The experimental outcomes are displayed in Tables 2 and 3.

**Table 2. Performance of bot detection using followers relationship graph.** Acc and F1 stands for accuracy and F1-score (macro), respectively. Boldface letters are used to mark the best results.

| Method | MGTAB | | Cresci-15 | | Twibot-20 | |
|---|---|---|---|---|---|---|
| Metric | Acc | F1 | Acc | F1 | Acc | F1 |
| GCN | 80.40 ± 0.75 | 70.16 ± 1.92 | 95.39 ± 0.38 | 95.02 ± 0.32 | 70.13 ± 0.47 | 69.78 ± 0.66 |
| SAGE | 81.10 ± 0.80 | 76.79 ± 1.15 | 94.87 ± 0.31 | 94.46 ± 0.35 | 73.13 ± 0.67 | 72.72 ± 0.67 |
| GAT | 81.85 ± 0.64 | 75.38 ± 0.93 | 95.21 ± 0.39 | 94.79 ± 0.43 | 74.63 ± 0.96 | 74.16 ± 1.04 |
| JK-Nets | 83.28 ± 0.19 | 76.99 ± 0.73 | 93.82 ± 1.60 | 93.42 ± 1.59 | 70.49 ± 0.70 | 70.04 ± 0.88 |
| LA-GCN | 85.20 ± 0.25 | 80.81 ± 0.35 | 95.24 ± 0.54 | 95.86 ± 0.49 | 73.72 ± 0.72 | 72.87 ± 0.69 |
| RF-GNN | 85.90 ± 0.95 | 81.29 ± 1.21 | 95.40 ± 0.47 | 95.03 ± 0.51 | 80.08 ± 0.26 | 79.70 ± 0.23 |
| DropEdge+GCN | 80.78 ± 0.47 | 71.48 ± 1.65 | 95.53 ± 0.41 | 95.18 ± 0.43 | 69.65 ± 0.77 | 69.36 ± 0.71 |
| DropEdge+SAGE | 81.03 ± 0.85 | 76.67 ± 1.12 | 94.78 ± 0.57 | 94.36 ± 0.62 | 75.97 ± 0.66 | 75.62 ± 0.68 |
| DropEdge+GAT | 81.34 ± 0.50 | 74.23 ± 1.06 | 95.34 ± 0.37 | 94.94 ± 0.40 | 72.70 ± 1.52 | 72.27 ± 1.54 |
| MixupForGraph+GCN | 85.73 ± 0.54 | 81.33 ± 1.45 | 95.47 ± 0.22 | 95.16 ± 0.23 | 77.91 ± 0.43 | 77.50 ± 0.46 |
| MixupForGraph+SAGE | 85.91 ± 0.70 | 81.96 ± 1.18 | 95.81 ± 0.36 | 95.49 ± 0.37 | 78.94 ± 0.62 | 78.57 ± 0.63 |
| MixupForGraph+GAT | 85.46 ± 0.62 | 81.03 ± 0.43 | 95.75 ± 0.22 | 95.43 ± 0.23 | 78.19 ± 0.34 | 77.72 ± 0.29 |
| G-CutMix(GCN) | 86.54 ± 0.73 | 82.81 ± 1.20 | 95.61 ± 0.21 | 95.26 ± 0.25 | **80.27 ± 0.59** | **79.86 ± 0.51** |
| G-CutMix(SAGE) | 85.97 ± 0.92 | 82.13 ± 1.12 | 95.99 ± 0.11 | 95.64 ± 0.12 | 79.44 ± 0.59 | 78.97 ± 0.53 |
| G-CutMix(GAT) | **86.61 ± 0.67** | **82.86 ± 1.19** | **96.05 ± 0.49** | **95.74 ± 0.53** | 80.23 ± 0.57 | 79.74 ± 0.54 |

**Table 3. Performance of bot detection using friends relationship graph.** Acc and F1 stands for accuracy and F1-score (macro), respectively. Boldface letters are used to mark the best results.

| Method | MGTAB | | Cresci-15 | | Twibot-20 | |
|---|---|---|---|---|---|---|
| Metric | Acc | F1 | Acc | F1 | Acc | F1 |
| GCN | 78.89 ± 0.53 | 65.28 ± 0.77 | 95.29 ± 0.35 | 95.13 ± 0.41 | 67.15 ± 0.73 | 64.65 ± 1.47 |
| SAGE | 79.59 ± 0.42 | 67.83 ± 1.13 | 94.87 ± 0.31 | 94.46 ± 0.35 | 66.52 ± 0.64 | 64.15 ± 1.08 |
| GAT | 80.04 ± 0.35 | 73.86 ± 0.66 | 95.21 ± 0.39 | 94.79 ± 0.43 | 68.47 ± 1.18 | 66.65 ± 1.42 |
| JK-Nets | 84.86 ± 1.77 | 78.96 ± 3.68 | 94.54 ± 1.42 | 94.16 ± 1.43 | 72.71 ± 0.54 | 72.48 ± 0.43 |
| LA-GCN | 85.34 ± 0.75 | 79.53 ± 1.32 | 95.22 ± 1.23 | 94.32 ± 1.02 | 74.52 ± 0.63 | 74.72 ± 0.55 |
| RF-GNN | 85.90 ± 0.95 | 81.29 ± 1.21 | 95.40 ± 0.47 | 95.03 ± 0.51 | 80.08 ± 0.26 | 79.70 ± 0.23 |
| DropEdge+GCN | 82.34 ± 0.62 | 74.48 ± 1.50 | 95.65 ± 0.42 | 95.29 ± 0.44 | 75.42 ± 0.34 | 74.65 ± 0.47 |
| DropEdge+SAGE | 86.15 ± 0.55 | 81.74 ± 0.82 | 94.78 ± 0.57 | 94.36 ± 0.62 | 77.74 ± 0.32 | 77.32 ± 0.31 |
| DropEdge+GAT | 85.16 ± 0.34 | 80.44 ± 0.81 | 95.34 ± 0.37 | 94.94 ± 0.39 | 75.42 ± 0.34 | 74.65 ± 0.47 |
| MixupForGraph+GCN | 86.17 ± 0.61 | 81.65 ± 0.82 | 95.93 ± 0.24 | 95.62 ± 0.25 | 75.21 ± 0.52 | 74.38 ± 0.74 |
| MixupForGraph+SAGE | 85.97 ± 0.46 | 82.01 ± 0.69 | 95.79 ± 0.28 | 95.45 ± 0.31 | 77.95 ± 0.68 | 77.43 ± 0.76 |
| MixupForGraph+GAT | 86.18 ± 0.51 | 81.88 ± 0.88 | 95.67 ± 0.32 | 95.33 ± 0.33 | 79.17 ± 0.59 | 78.57 ± 0.52 |
| G-CutMix(GCN) | **86.67 ± 0.84** | **82.99 ± 1.45** | 96.01 ± 0.35 | 95.69 ± 0.38 | 79.17 ± 0.58 | 78.74 ± 0.60 |
| G-CutMix(SAGE) | 86.92 ± 0.49 | 82.78 ± 0.68 | 96.02 ± 0.12 | 95.72 ± 0.13 | 78.08 ± 0.70 | 77.53 ± 0.90 |
| G-CutMix(GAT) | 86.08 ± 0.42 | 81.98 ± 0.37 | **96.09 ± 0.34** | **95.80 ± 0.37** | **79.65 ± 0.83** | **79.18 ± 0.79** |

Utilizing both followers relationship graphs and friends relationship graphs, our proposed G-CutMix for bots detection consistently outperforms baseline methods. Employing GCN, SAGE, and GAT as backbone models, and comparing with graph data augmentation techniques like DropEdge and MixupForGraph, our G-CutMix, through multi-level data enhancement, achieves superior performance.

Randomly selecting 1,000 nodes, the t-SNE dimensionality reduction and visualization of the hidden layer features obtained by various methods are depicted in Fig 3, with labels 1 and 0 representing bots and human accounts, respectively. The embeddings obtained by the GCN model exhibit the highest degree of overlap in the feature space, as evidenced by the fact that most green points are occluded by orange points. DropEdge and MixupGCN demonstrate better distinguishability in the feature space compared to GCN. The embeddings generated by CutMix show the lowest overlap in the feature space and achieve the highest distinguishability. In the case of the GCN model, the model's representational capacity is insufficient, resulting in a significant overlap of the hidden layer features between bots and human accounts, which indicates poor separability. DropEdge randomly discards edges, but the effect on feature enhancement is not pronounced. In contrast, G-CutMix and MixupForGraph enhance the separability of features through data augmentation of node features. G-CutMix outperforms both DropEdge and MixupForGraph in terms of effectiveness.

## Discussion

### Training set sizing

In the field of bots detection, obtaining a large-scale labeled dataset is often challenging. In this section, we showcase that our G-CutMix approach results in more significant improvements when trained on smaller datasets. The validation set and test set are the same as CutMix for Heterogeneous Graph section, changing the training set size from 1% to 5%. The results are reported in Fig 4.

Upon examining the performance of G-CutMix and the original GNN, it is evident that our G-CutMix approach enhances the performance of GCN and GAT across various training

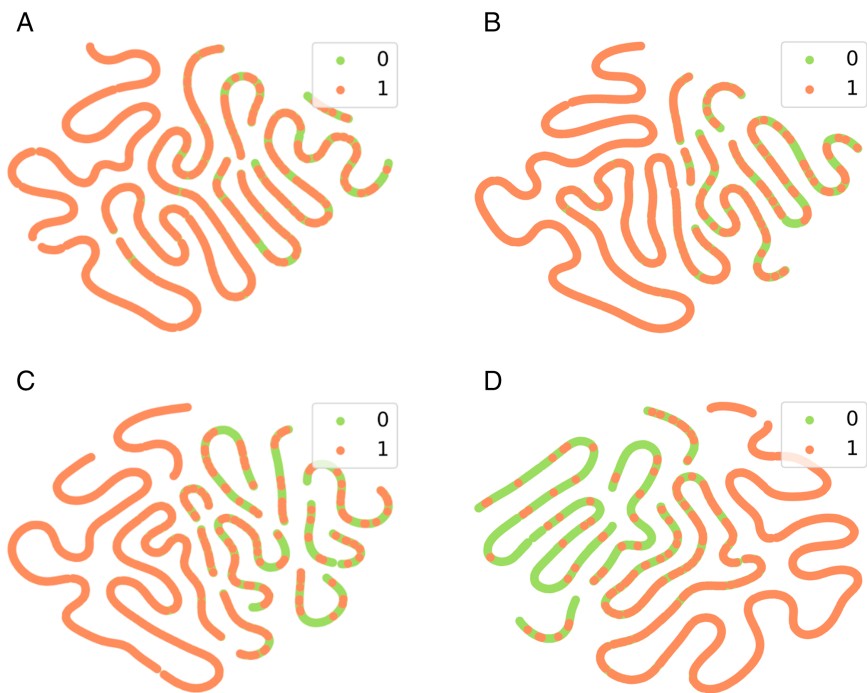

**Fig 3. The t-SNE visualizations of the output results from various methods.** (**a**) GCN. (**b**) DropEdge. (**c**) MixupForGrap. (**d**) CutMix.

set sizes. Our method shows more significant improvements for smaller training sets because it is challenging for the model to achieve sufficient training with limited training data.

## Parameters sensitivity analysis

The parameter $\alpha$ controls the CutMix ratio, and in this section, we analyze the effect of $\alpha$ in our method. When $\alpha$ equals 0, the node features of the two graph views are not augmented with CutMix. When $a$ equals 1, the mix ratio of features in $\mathcal{G}_A$ and $\mathcal{G}_B$ is maximized, with features directly exchanged.

Experiments in Fig 5 demonstrate that $\alpha = 0.3$ achieves optimal performance across all datasets (MGTAB, Cresci-15, Twibot-20). Specifically, varying $\alpha$ from 0.1 to 0.9 on MGTAB results in accuracy fluctuations within $\pm 1.5\%$, suggesting parameter robustness. We further propose empirical guidelines: datasets with sparse graphs (e.g., Twibot-20) achieve best performance with $\alpha \in [0.2, 0.4]$, whereas $\alpha \in [0.3, 0.5]$ is preferred for dense graphs like MGTAB.

## CutMix for heterogeneous graph

In the field of social bots detection, it has been observed that using multiple relations instead of just a single relation leads to better performance. Our proposed approach can be extended to multi-relation graphs, where multiple relations are used simultaneously. To this end, we constructed a social network graph with two relations, namely followers and friends, using a machine detection dataset. The comparison results of G-CutMix with classic heterogeneous graph neural networks RGCN [37] and RGAT [29] are presented in Table 4. This approach

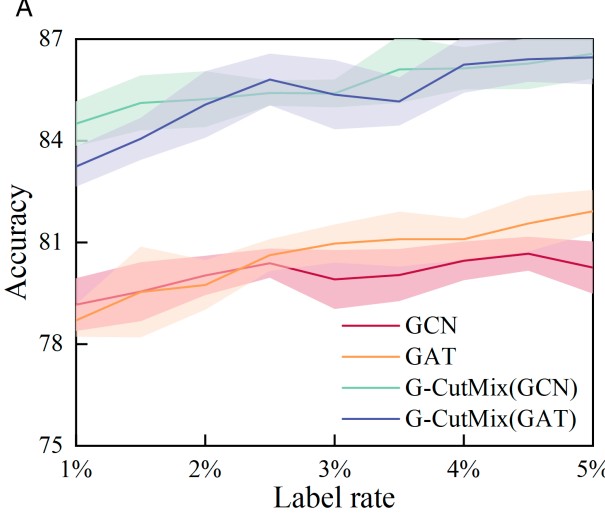

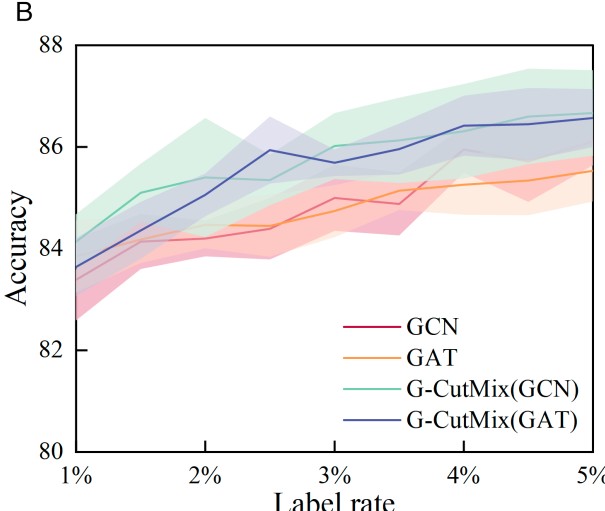

**Fig 4. Accuracy of different training sets using different relationship graph.** (**a**) Results based on friends relationship. (**b**) Results based on followers relationship.

proves to be advantageous since G-CutMix enables the model to generalize better to unseen data by leveraging information from multiple relations.

The consistent superiority of G-CutMix-enhanced models, average 1.91% F1 improvements on MGTAB, 1.95% on Cresci-15, 3.82% on Twibot-20 stems from its ability to synergize follower and friend relations. Unlike baseline RGCN/RGAT that process relations sequentially, our method's isomorphic shuffling and feature fusion create implicit cross-relational attention—for instance, amplifying signals where follower-friend reciprocity indicates coordinated bot behavior. While both RGCN and RGAT benefit from G-CutMix, the greater improvements with RGAT highlight our method's compatibility with attention mechanisms. The learnable merging weights in G-CutMix likely synergize with RGAT's edge-specific attention, enabling adaptive reweighting of mixed features.

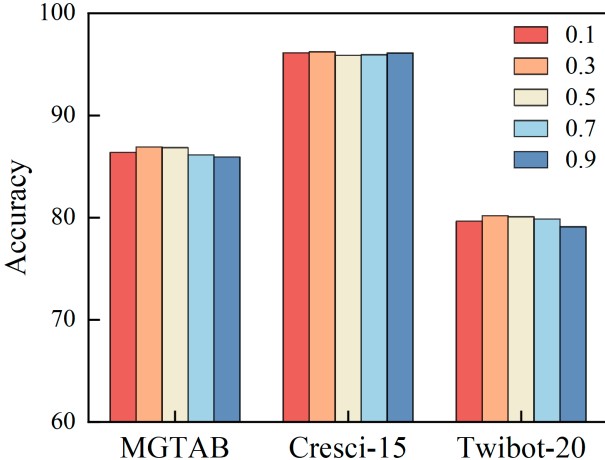

**Fig 5. Performance of G-CutMix with different $\alpha$.**

**Table 4**. **Performance on multi-relation graphs using both friendship and follower relations.** Boldface letters indicate the best results.

| Method | MGTAB | | Cresci-15 | | Twibot-20 | |
|---|---|---|---|---|---|---|
| Metric | Acc | F1 | Acc | F1 | Acc | F1 |
| RGCN | 86.06 ± 0.83 | 81.54 ± 1.15 | 94.13 ± 0.47 | 93.65 ± 0.48 | 75.68 ± 1.04 | 75.27 ± 1.03 |
| RGAT | 86.23 ± 0.90 | 81.69 ± 1.03 | 94.13 ± 0.42 | 93.89 ± 0.44 | 76.73 ± 0.82 | 76.22 ± 0.92 |
| G-CutMix+RGCN | **86.96 ± 1.00** | **83.40 ± 1.51** | 96.07 ± 0.43 | 95.76 ± 0.45 | 80.27 ± 0.73 | 79.82 ± 0.71 |
| G-CutMix+RGAT | 86.76 ± 0.79 | 83.20 ± 1.15 | **96.10 ± 0.47** | **95.80 ± 0.50** | **80.69 ± 0.77** | **80.09 ± 0.79** |

## Ablation study

We divided the dataset into training, validation, and test sets as outlined in Main Results. To evaluate the impact of each module in G-CutMix, we used the followers relationship graph. Specifically, we conducted experiments by removing the Node Shuffle Module, Graph Cut-Mix Module, and Attribute Connection Module individually. The settings "w/o shuffle", "w/o CutMix", and "w/o attribute" correspond to our original method without the Node Shuffle Module, Graph CutMix Module, and Attribute Connection Module, respectively.

To investigate the effects of node shuffle and CutMix in our method, we removed the Node Shuffle Module and set the mixing parameter $\alpha$ in CutMix to be 0. The results of these experiments are presented in Table 5.

The ablation study results in Table 5 critical interdependencies between G-CutMix's components and dataset characteristics. The most pronounced performance degradation occurs when removing the Attribute Connection Module (average 3.8% F1 drop across datasets), particularly severe in Twibot-20 (6.2% accuracy decline for GCN), suggesting that social bots' attribute camouflage strategies - such as profile metadata manipulation - require explicit attribute correlation modeling to detect. Interestingly, while Node Shuffle removal impacts MGTAB most significantly (1.5-2.5% accuracy reduction), its effect diminishes in Twibot-20 where temporal behavioral patterns dominate, implying that structural isomorphism preservation becomes less critical when bots exhibit strong activity sequence signatures. These findings collectively demonstrate that G-CutMix's power emerges from the synergistic combination of its components rather than any single module.

**Table 5**. **The accuracy of G-CutMix and its variants.** Boldface letters are used to mark the best results.

|  | Methods | Origin | w/o shuffle | w/o CutMix | w/o attributes |
|---|---|---|---|---|---|
| MGTAB | G-CutMix(GCN) | **86.54 ± 0.73** | 84.52 ± 0.87 | 85.46 ± 1.00 | 83.02 ± 0.81 |
|  | G-CutMix(SAGE) | **85.97 ± 0.92** | 85.49 ± 0.96 | 85.45 ± 0.82 | 85.41 ± 0.80 |
|  | G-CutMix(GAT) | **86.61 ± 0.67** | 85.96 ± 1.16 | 85.59 ± 0.94 | 83.14 ± 0.66 |
| Cresci-15 | G-CutMix(GCN) | **95.61 ± 0.21** | 94.66 ± 0.52 | 94.83 ± 0.47 | 95.24 ± 0.34 |
|  | G-CutMix(SAGE) | **95.99 ± 0.11** | 94.66 ± 0.52 | 94.22 ± 0.56 | 95.72 ± 0.39 |
|  | G-CutMix(GAT) | **96.05 ± 0.49** | 94.73 ± 0.54 | 94.70 ± 0.60 | 95.72 ± 0.39 |
| Twibot-20 | G-CutMix(GCN) | **79.17 ± 0.58** | 79.06 ± 0.67 | 79.39 ± 0.52 | 72.43 ± 0.80 |
|  | G-CutMix(SAGE) | **78.08 ± 0.70** | 78.34 ± 0.74 | 77.92 ± 0.78 | 76.99 ± 0.76 |
|  | G-CutMix(GAT) | **79.65 ± 0.83** | 78.81 ± 0.90 | 78.73 ± 0.77 | 74.63 ± 0.43 |

**Table 6**. **Computational complexity comparisons for different methods.**

| Method | MGTAB | Twibot-20 | Cresci-15 |
|---|---|---|---|
| GCN | 16.11 | 18.92 | 9.35 |
| SAGE | 18.64 | 21.56 | 10.13 |
| GAT | 20.53 | 24.58 | 12.62 |
| G-CutMix (GCN) | 20.84 | 22.54 | 11.53 |
| G-CutMix (SAGE) | 22.36 | 24.84 | 12.43 |
| G-CutMix (GAT) | 23.62 | 27.15 | 12.98 |

## Computational complexity

On MGTAB with a GCN backbone, G-CutMix increases training time by 20% compared to vanilla GCN, as shown in Table 6. These results confirm that G-CutMix's performance gains outweigh its modest computational cost, making it suitable for real-world deployment.

## Conclusions

In this paper, we introduce a novel graph data augmentation method based on CutMix to enhance the performance of bots detection in social networks. Our approach involves mixing the node features and labels of two graphs in sequence using CutMix, followed by feeding the mixed features into a graph convolution. The convolution generates local aggregation-based features, which are then combined with the original features of the two graphs to produce a new layer of features. Through extensive experiments, we demonstrate the effectiveness of our proposed approach, G-CutMix, in detecting bots on social networks, especially in scenarios with limited labeled nodes.

## Author contributions

**Conceptualization:** Yan Li.

**Formal analysis:** Xiaofeng Guo.

**Funding acquisition:** Xiaofeng Guo.

**Investigation:** Shuhao Shi, Xiaofeng Guo.

**Methodology:** Shuhao Shi.

**Project administration:** Chunhua Zhou.

**Resources:** Chunhua Zhou.

**Software:** Qian Hu.

**Validation:** Qian Hu.

**Writing – original draft:** Yan Li.

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
