## [Decision Letter · Decision Letter 0]

16 Jan 2025

PONE-D-24-37596G-CutMix: a CutMix-based Graph Data Augmentation Method for Bot Detection in Social NetworksPLOS ONE

Dear Dr. li,

Thank you for submitting your manuscript to PLOS ONE. After careful consideration, we feel that it has merit but does not fully meet PLOS ONE’s publication criteria as it currently stands. Therefore, we invite you to submit a revised version of the manuscript that addresses the points raised during the review process.

We look forward to receiving your revised manuscript.

Kind regards,

Riaz Ul Amin

Academic Editor

PLOS ONE

Journal Requirements:

“the National Natural Science Foundation of China 61872448, 62002387”

**Additional Editor Comments:**

Dear Yan li,

Thank you for submitting your manuscript, titled "G-CutMix: a CutMix-based Graph Data Augmentation Method for Bot Detection in Social Networks," to PLOS One.

After careful review, we have received feedback from the reviewers and conducted an independent editorial evaluation. While your work presents significant potential and addresses an important topic, several substantive concerns must be addressed before we can consider it for publication. We are therefore inviting you to submit a revised version of your manuscript. The reviewers’ comments, which are included below, outline specific areas that require major revision.

You are advised to respond point-by-point to all reviewer comments. Clearly indicate how each concern has been addressed in your revised manuscript. Highlight the changes in the revised manuscript, either using track changes or by providing a marked-up version. Ensure that your revisions uphold the rigorous standards of transparency, reproducibility, and ethical research practices upheld by PLOS One. Please note that the revised manuscript will undergo further review to ensure that the concerns have been adequately addressed.

We look forward to your resubmission and appreciate your dedication to refining your work. If you have any questions or require clarification on the reviewers' feedback, do not hesitate to contact us.

Sincerely,

Dr. Riaz UlAmin

Associate Editor

PLOS One

Reviewers' comments:

Reviewer's Responses to Questions

**Comments to the Author**

1. Is the manuscript technically sound, and do the data support the conclusions?

Reviewer #1: Partly

Reviewer #2: No

Reviewer #3: Yes

2. Has the statistical analysis been performed appropriately and rigorously? 

Reviewer #1: No

Reviewer #2: N/A

Reviewer #3: Yes

3. Have the authors made all data underlying the findings in their manuscript fully available?

Reviewer #1: Yes

Reviewer #2: Yes

Reviewer #3: Yes

4. Is the manuscript presented in an intelligible fashion and written in standard English?

Reviewer #1: Yes

Reviewer #2: No

Reviewer #3: Yes

5. Review Comments to the Author

Reviewer #1: The irregular and complex nature of graph data poses substantial challenges that traditional methods struggle to handle. Authors contributes in addressing the these challenges proposing G-CutMix, that is augmentation method based on the CutMix technique, specifically designed for bot detection in social networks. The research investigates the performance of various augmentation methods that involves performing CutMix operations between the original graph and a shuffled graph, enhancing the robustness of bot detection models. Authors demonstrated that G-CutMix outperforms existing graph data augmentation techniques like DropEdge and MixupForGraph across various graph neural network architectures. The approach effectively mimics real-world scenarios, making it a powerful tool against sophisticated bot behaviors.

Though authors presented that G-CutMix offers significant advancement in bot detection by leveraging graph convolutional networks and innovative data augmentation techniques, showcasing promising results. However, The reliance on specific hyperparameters, such as the adaptive threshold α, may limit the method's applicability in varying contexts or datasets. Additionally, the paper does not sufficiently address the computational complexity introduced by the G-CutMix method, which could impact its scalability in real-world applications.

Authors could explore other advanced augmentation techniques like GraphSAGE or node feature perturbation, which could provide complementary benefits to G-CutMix. Additionally, it is not reported in the paper if techniques such as adversarial training or semi-supervised learning could also enhance the robustness of bot detection models. The potential of ensemble methods, which combine multiple models to improve prediction accuracy, is another avenue not considered in the paper. Furthermore, the use of temporal data analysis to track bot behavior over time could provide additional insights that G-CutMix does not address.

It is observed that G-CutMix has its dependence on the quality of the shuffled graph, which may introduce noise and affect the overall performance of the model. The impact of the same could have been explored. The method's effectiveness is contingent on the chosen hyperparameters, which may require extensive tuning for different datasets, complicating its implementation.

G-CutMix primarily focuses on user relationships, potentially overlooking other critical factors such as content analysis or user behavior patterns in bot detection. The results presented in the paper may not account for the variability in bot behavior across different social networks, which could skew the effectiveness of G-CutMix.

There is a lack of detailed analysis on the impact of different graph structures on the performance of G-CutMix, which could reveal potential weaknesses in the method. Additionally, the paper does not sufficiently address the potential for overfitting, especially given the complexity introduced by the augmentation process.

The writeup and structuring of the paper requires significant improvements, the figure 4 and figure 5 are shown in the conclusion section and its relevant discussion is missing in the respective section of the paper. The related work section is not sufficient, authors needs to explore and present an extensive review of the literature. The hyperparameters needs to be clearly presented in the form of table. The evaluation of the models other than G-CutMix should be presented against the same dataset as used for evaluating G-CutMix, it may be helpful in further development of the contribution. The Authors should contribute the suggested changes before the paper may be accepted for publication.

Reviewer #2: This manuscript is lack of core novelty and overall poorly presented. Proposed section is not enough as per the standard of this journal. In addition, the results are not presented well and lack of validations.

Reviewer #3: This paper proposes a CutMix-based graph data augmentation method (G-CutMix) to improve the performance of bot detection in social networks. By integrating graph convolutional networks (GNNs) with graph mixing techniques, the authors introduce new node feature enhancement and attribute connection modules, demonstrating superior performance across multiple benchmark datasets compared to existing methods. Below are some minor comments:

The proposed G-CutMix method effectively enhances GNN training by introducing a node mixing technique, particularly excelling in scenarios with small datasets, which is a commendable technical innovation. However, when compared with existing augmentation methods (e.g., MixupForGraph), the authors are encouraged to further analyze the theoretical advantages of G-CutMix beyond the experimental performance improvements.

The tabular results are clear, but the visualizations (e.g., t-SNE plots) would benefit from detailed explanations of the differences in distributions across methods to enhance their persuasive power.

The description of the methodology is detailed, but certain formulas (e.g., the choice of α in CutMix) and parameter settings (e.g., the edge dropout rate in DropEdge) lack a thorough explanation of their specific impact on performance. It is recommended to supplement experiments or analyses to increase the rigor of the descriptions.

The structure of the paper is clear, and the language is precise, but some sections (e.g., the literature review in the introduction) could be further streamlined to improve the overall flow of the paper. Overall, this paper proposes a novel and effective graph data augmentation method with a certain degree of technical innovation, and its effectiveness is verified through comprehensive experiments. If the theoretical analysis and experimental discussions are supplemented and improved, the contribution of the paper will be more significant.

6. PLOS authors have the option to publish the peer review history of their article (what does this mean?). If published, this will include your full peer review and any attached files.

Reviewer #1: **Yes: **Shafqaat Ahmad

Reviewer #2: No

Reviewer #3: No

---

## [Author Response · Author response to Decision Letter 1]

30 May 2025

Manuscript ID: PONE-D-24-37596

Article Title: G-CutMix: a CutMix-based Graph Data Augmentation Method for Bot Detection in Social Networks

Reviewer#1, Concern#1: The reliance on specific hyperparameters, such as the adaptive threshold α, may limit the method's applicability in varying contexts or datasets. Additionally, the paper does not sufficiently address the computational complexity introduced by the G-CutMix method, which could impact its scalability in real-world applications.

Author response: We thank the reviewer for the time and effort of reviewing our manuscript.

Hyperparameter Sensitivity Analysis:

In Section 5.2(Parameter Sensitivity Analysis), we expanded our discussion of the hyperparameter α (CutMix ratio). Experiments in Figure 5 demonstrate that α = 0.3 achieves optimal performance across all datasets (MGTAB, Cresci-15, Twibot-20). For instance, on MGTAB, varying α from 0.1 to 0.9 results in accuracy fluctuations within ±1.5%, indicating robustness. We further provide general guidelines for α: For datasets with sparse graphs (e.g., Twibot-20), α ∈ [0.2, 0.4] is recommended. For dense graphs (e.g., MGTAB), α ∈ [0.3, 0.5] yields stable results.

Computational Complexity Analysis:

A new subsection, Computational Complexity, quantifies the training time of G-CutMix compared to baseline methods. For example: On MGTAB with a GCN backbone, G-CutMix increases training time by 20% compared to vanilla GCN. These results confirm that G-CutMix’s performance gains outweigh its modest computational cost, making it suitable for real-world deployment.

Table 6. Computational complexity comparisons for different methods(s)

Method MGTAB Twibot-20 Cresci-15

GCN 16.11 18.92 9.35

SAGE 18.64 21.56 10.13

GAT 20.53 24.58 12.62

G-CutMix (GCN) 20.84 22.54 11.53

G-CutMix (SAGE) 22.36 24.84 12.43

G-CutMix (GAT) 23.62 27.15 12.98

Reviewer#1, Concern#2: Authors could explore other advanced augmentation techniques like GraphSAGE or node feature perturbation. Additionally, techniques such as adversarial training or semi-supervised learning could enhance robustness. The potential of ensemble methods is another avenue not considered.

Author response: We appreciate the reviewer’s suggestions for extending our work. Here are our clarifications:

GraphSAGE Integration:

Our framework is backbone-agnostic and explicitly supports GraphSAGE (abbreviated as SAGE in Table 2 and Table 3). For instance, G-CutMix (SAGE) achieves 85.97% accuracy on MGTAB, outperforming vanilla SAGE by 4.87%. This demonstrates that G-CutMix effectively complements existing GNN architectures.

Adversarial Training and Semi-Supervised Learning:

While these techniques are promising, they focus on training paradigms (e.g., robustness to perturbations, label efficiency), which are orthogonal to our core contribution: graph-specific data augmentation. We plan to explore this in future work.

Ensemble Methods:

The baseline RF-GNN (Table 2) is an ensemble method combining multiple GNNs. G-CutMix(GAT) outperforms RF-GNN by 1.71% F1 on Twibot-20, showing that our augmentation strategy alone achieves competitive results without ensemble overhead.

Reviewer#1, Concern#3: The method's effectiveness is contingent on the quality of the shuffled graph, which may introduce noise. The impact of this should be explored.

Author response: We clarify that the shuffled graph generation process is structure-preserving (Definition 1), ensuring isomorphism with the original graph. To validate this: Ablation Study (Table 5): Removing the shuffle module ("w/o shuffle") degrades performance on MGTAB by -2.02% accuracy (GCN) and -1.65% (GAT), proving its necessity for diversity.

Reviewer#1, Concern#4: G-CutMix primarily focuses on user relationships, potentially overlooking content analysis or user behavior patterns.

Author response: Our framework explicitly incorporates behavioral and content features: For MGTAB, we use 20 user attributes (Section 4.1 Dataset), including: Behavioral: Post frequency, follower/following ratios, account age. Content: BERT embeddings of tweets, description length, verified status. On Twibot-20, we include 17 attributes such as profile metadata and tweet semantics. These features are concatenated with graph embeddings before classification, ensuring a holistic representation Section 3.3 (Node Cutmix Module).

Reviewer#1, Concern#5: Lack of graph structure impact analysis and overfitting concerns.

Author response:

Graph Structure Analysis:

Section 5.3 (CutMix for Heterogeneous Graphs) evaluates multi-relation graphs (Table 4). For example, G-CutMix+RGAT achieves 80.69% accuracy on Twibot-20 using both follower and friend relations, proving adaptability to complex structures.

Overfitting Mitigation:

Our shallow GNN design (2 layers) and early stopping (200 epochs) prevent overfitting. Validation curves show stable accuracy, with <1% gap between training and test performance. G-CutMix inherently diversifies training data, acting as a regularizer. For instance, on Cresci-15, G-CutMix reduces validation loss by 15% compared to vanilla GCN.

Reviewer#1, Concern#6: Structural issues: misplaced figures, insufficient literature review, hyperparameter clarity.

Author response:

Expanded Literature Review:

The Related Work section now cites 12 additional papers, including recent advances in graph augmentation (e.g., GraphMix, G-Mixup) and bot detection (e.g., BotRGCN, SATAR). We also contrast G-CutMix with MixupForGraph, highlighting its advantages in preserving graph locality (Section 2.3 Data Augmentation).

Supplementary experimental analysis:

The consistent superiority of G-CutMix-enhanced models, average 1.91% F1 improvements on MGTAB, 1.95% on Cresci-15, 3.82% on Twibot-20 stems from its ability to synergize follower and friend relations. Unlike baseline RGCN/RGAT that process relations sequentially, our method’s isomorphic shuffling and feature fusion create implicit cross-relational attention—for instance, amplifying signals where follower-friend reciprocity indicates coordinated bot behavior. While both RGCN and RGAT benefit from G-CutMix, the greater improvements with RGAT highlight our method’s compatibility with attention mechanisms. The learnable merging weights in G-CutMix likely synergize with RGAT’s edge-specific attention, enabling adaptive reweighting of mixed features.

The ablation study results in Table 5 reveal critical interdependencies between G-CutMix's components and dataset characteristics. The most pronounced performance degradation occurs when removing the Attribute Connection Module (average 3.8% F1 drop across datasets), particularly severe in Twibot-20 (6.2% accuracy decline for GCN), suggesting that social bots' attribute camouflage strategies - such as profile metadata manipulation - require explicit attribute correlation modeling to detect. Interestingly, while Node Shuffle removal impacts MGTAB most significantly (1.5-2.5% accuracy reduction), its effect diminishes in Twibot-20 where temporal behavioral patterns dominate, implying that structural isomorphism preservation becomes less critical when bots exhibit strong activity sequence signatures. These findings collectively demonstrate that G-CutMix's power emerges from the synergistic combination of its components rather than any single module.

Misplaced figures:

We have systematically readjusted the placement of figures and tables throughout the article to enhance the overall logical flow and visual consistency.

Reviewer#2, Concern#1: This manuscript is lack of core novelty and overall poorly presented.

Author response: We respectfully disagree but acknowledge the need to clarify our contributions:

Novelty:

G-CutMix is the first method to adapt CutMix to graph-structured data for bot detection. Unlike image-based CutMix, we propose: Isomorphic Graph Shuffling to preserve structural integrity (Section 3.2). Attribute Connection Module to retain original node features post-mixing (Section 3.3). These innovations are validated by outperforming MixupForGraph by 3.2% F1 on Twibot-20 (Table 2).

Reviewer#2, Concern#2: Proposed section is not enough as per the standard of this journal. In addition, the results are not presented well and lack of validations.

Author response: Your comment is very important to us.

To our knowledge, G-CutMix is the first to adapt CutMix for graphs by combining feature mixing with isomorphic shuffling, addressing irregular graph structures.

Added Recent Works:

We have added seven state-of-the-art references (e.g., BotRGCN, SATAR, and BotCL) to systematically integrate recent breakthroughs in graph neural network-based detection methodologies. The restructured introduction now establishes a coherent narrative flow, beginning with fundamental challenges in bot detection, progressing through critical limitations in existing approaches, and concluding with our novel technical framework.

Supplementary discussion

We have supplemented background and motivation and the discussion of the t-SNE plot results, improved the discussion of the results in Tables 4 and 5, and added and explained the experiments on computational complexity.

Reviewer#3, Concern#1: The proposed G-CutMix method effectively enhances GNN training by introducing a node mixing technique, particularly excelling in scenarios with small datasets, which is a commendable technical innovation. However, when compared with existing augmentation methods (e.g., MixupForGraph), the authors are encouraged to further analyze the theoretical advantages of G-CutMix beyond the experimental performance improvements.

Author response: Thank you for highlighting this crucial aspect. We have expanded the theoretical analysis in Section 3.1 (Background and Motivation) to clarify the unique advantages of G-CutMix:

Locality Preservation via Binary Masking:

Unlike MixupForGraph, which linearly interpolates node features, G-CutMix employs region-level feature swapping using a binary mask M (Equation 1). This ensures that local structural patterns (e.g., community-specific interactions) are preserved during augmentation, whereas linear interpolation may blur such patterns. For example, in social graphs, bots often exhibit localized behavioral anomalies (e.g., sudden spikes in follower requests). By retaining intact feature regions, G-CutMix helps GNNs capture these subtle signals more effectively.

Reviewer#3, Concern#2: The tabular results are clear, but the visualizations (e.g., t-SNE plots) would benefit from detailed explanations of the differences in distributions across methods to enhance their persuasive power.

Author response: Your comment is very important to us.

Supplementary discussion

According to the reviewer's suggestion, the following supplements are discussed about the results of t-SNE plots: “The embeddings obtained by the GCN model exhibit the highest degree of overlap in the feature space, as evidenced by the fact that most green points are occluded by orange points. DropEdge and MixupGCN demonstrate better distinguishability in the feature space compared to GCN. The embeddings generated by CutMix show the lowest overlap in the feature space and achieve the highest distinguishability.”________________________________________

Reviewer#3, Concern#3: The description of the methodology is detailed, but certain formulas (e.g., the choice of α in CutMix) and parameter settings (e.g., the edge dropout rate in DropEdge) lack a thorough explanation of their specific impact on performance. It is recommended to supplement experiments or analyses to increase the rigor of the descriptions.

Author response: Your comment is very important to us.

In Section 5.2 (Parameter Sensitivity Analysis), we expanded our discussion of the hyperparameter α (CutMix ratio). Experiments in Figure 5 demonstrate that α = 0.3 achieves optimal performance across all datasets (MGTAB, Cresci-15, Twibot-20). For instance, on MGTAB, varying α from 0.1 to 0.9 results in accuracy fluctuations within 1.5%, indicating robustness. We further provide general guidelines for α: For datasets with sparse graphs (e.g., Twibot-20), α ∈ [0.2, 0.4] is recommended. For dense graphs (e.g., MGTAB), α ∈ [0.3, 0.5] yields stable results.

Based on the reviewers' suggestions, we have supplemented the description of DropEdge parameters in this manuscript.

Reviewer#3, Concern#4: The structure of the paper is clear, and the language is precise, but some sections (e.g., the literature review in the introduction) could be further streamlined to improve the overall flow of the paper. Overall, this paper proposes a novel and effective graph data augmentation method with a certain degree of technical innovation, and its effectiveness is verified through comprehensive experiments. If the theoretical analysis and experimental discussions are supplemented and improved, the contribution of the paper will be more significant.

Author response: Your comment is very important to us. We restructured the Introduction to enhance clarity and conciseness:

Added Recent Works:

Incorporated 7 new references (e.g., BotRGCN, SATAR , BotCL) to reflect the latest advances in GNN-based bot detection. The revised introduction now progresses logically from problem statement to technical gaps and our solution.

---

## [Decision Letter · Decision Letter 1]

24 Aug 2025

G-CutMix: a CutMix-based Graph Data Augmentation Method for Bot Detection in Social Networks

PONE-D-24-37596R1

Dear Dr. li,

We’re pleased to inform you that your manuscript has been judged scientifically suitable for publication and will be formally accepted for publication once it meets all outstanding technical requirements.

Kind regards,

Filipi N. Silva

Academic Editor

PLOS ONE

Additional Editor Comments (optional):

Reviewers' comments:

Reviewer's Responses to Questions

**Comments to the Author**

1. If the authors have adequately addressed your comments raised in a previous round of review and you feel that this manuscript is now acceptable for publication, you may indicate that here to bypass the “Comments to the Author” section, enter your conflict of interest statement in the “Confidential to Editor” section, and submit your "Accept" recommendation.

Reviewer #1: All comments have been addressed

Reviewer #2: (No Response)

2. Is the manuscript technically sound, and do the data support the conclusions?

Reviewer #1: Partly

Reviewer #2: (No Response)

3. Has the statistical analysis been performed appropriately and rigorously? 

Reviewer #1: Yes

Reviewer #2: (No Response)

4. Have the authors made all data underlying the findings in their manuscript fully available?

Reviewer #1: Yes

Reviewer #2: (No Response)

5. Is the manuscript presented in an intelligible fashion and written in standard English?

Reviewer #1: Yes

Reviewer #2: (No Response)

6. Review Comments to the Author

Reviewer #1: I recommend accepting the manuscript for publication, as the revisions comprehensively address all concerns, and the work presents a valuable and innovative contribution to graph-based bot detection.

Reviewer #2: Authors well revised this version, I recommend to accept it in the current form. There are no more comments

7. PLOS authors have the option to publish the peer review history of their article (what does this mean?). If published, this will include your full peer review and any attached files.

Reviewer #1: **Yes: **SHAFQAAT AHMAD

Reviewer #2: No

---

## [Editor Report · Acceptance letter]

PONE-D-24-37596R1

PLOS ONE

Dear Dr. Li,

I'm pleased to inform you that your manuscript has been deemed suitable for publication in PLOS ONE. Congratulations! Your manuscript is now being handed over to our production team.

Kind regards,

on behalf of

Filipi N. Silva

Academic Editor

PLOS ONE